# Lack of Association between (AAT)n Polymorphism of the *CNR1* Gene Encoding the Cannabinoid Receptor (CB1) and Patient’s Quality of Life

**DOI:** 10.3390/genes13112046

**Published:** 2022-11-06

**Authors:** Anna Machoy-Mokrzyńska, Monika Rać, Alina Jurewicz, Violetta Dziedziejko, Krzysztof Safranow, Mateusz Kurzawski, Agnieszka Boroń, Arkadiusz Stefaniak, Katarzyna Leźnicka, Andrzej Bohatyrewicz, Monika Białecka

**Affiliations:** 1Department of Pharmacology, Pomeranian Medical University, Al. Powstańców Wielkopolskich 72 St., 70-111 Szczecin, Poland; 2Department of Biochemistry and Medical Chemistry, Pomeranian Medical University, Al. Powstańców Wielkopolskich 72 St., 70-111 Szczecin, Poland; 3Department of Orthopedics, Traumatology and Orthopedic Oncology, Pomeranian Medical University, Unii Lubelskiej St. 1, 71-252 Szczecin, Poland; 4Department of Clinical and Molecular Biochemistry, Pomeranian Medical University in Szczecin, Al. Powstańców Wielkopolskich 72 St., 70-111 Szczecin, Poland; 5Faculty of Physical Education, Gdansk University of Physical Education and Sport, K. Górskiego St. 1, 80-336 Gdansk, Poland

**Keywords:** microsatellite (AAT)n polymorphism of *CNR1* gene, postoperative pain, hip function

## Abstract

Genetic factors may predispose persons to decreased pain excitability. One of the interesting modulators affecting pain perception may be polymorphisms of the cannabinoid receptor type 1 (*CNR1*) gene. In this study, we examined the association between three-nucleotide repeats (AAT) polymorphism located in the 3′UTR non-translational region of *CNR1* and the patient’s quality of life after total hip arthroplasty. Our study examined the degree of pain sensation, hip function, and the patient’s performance at defined intervals after elective hip replacement due to degenerative changes. The study included 198 patients (128 women and 70 men). The average age was 67 years. PCR genotyping assay was used to identify the (AAT)n triplet repeat polymorphism in the *CNR1* gene. The (AAT)n repeat number was determined by sequencing using a standard sequencing protocol. Our study found no statistically significant association between the degree of pain, hip function, and the change in the degree of disability and the (AAT)n polymorphism in the *CNR1* gene, no statistically significant correlations between clinical symptoms, the patient’s age, and the number of AAT repeats, no association between the length of the allele and the degree of pain, hip function, and the change in disability.

## 1. Introduction

Osteoarthritis (OA) is one of the more common causes of disability in the elderly. Pain and progressive loss of function are the most important clinical signs warranting the initiation of both pharmacological and surgical treatment [1,2]. The risk factors for osteoarthritis include female sex [3], obesity [4], congenital or acquired defect in joint structure or function [5], certain systemic diseases [6], occupation, e.g., farmers, professional runners [7], and genetic factors—mutations in genes encoding collagen fibers (e.g., COL11A1), other cartilage matrix proteins (e.g., COMP), vitamin D receptor (VDR) and estrogen receptor (ESR1), as well as growth factors present in bone and cartilage (VEGF) and cytokines (IL-1B, IL-6) [8,9,10,11,12,13,14]. In the pathophysiology of osteoarthritis, attention is drawn to the role of inflammatory processes in the synovial membrane, which can occur at any stage of the disease and cause pain of a different origin and location [15]. The main symptom of the disease is subjective in nature, and its perception in humans is modulated by many factors, such as gender, age, current mental and physical state, previous experiences, background, and genetic variability of the person experiencing the sensation of pain [16]. The first choice for pharmacological treatment of mild to moderate pain in osteoarthritis is paracetamol (acetaminophen) [17]. Non-steroidal anti-inflammatory drugs (NSAIDs), on the other hand, should be introduced when treatment with paracetamol is not satisfactory [18]. Intra-articular corticosteroid injections can also serve to treat moderate to severe pain when non-opioid analgesics fail to reduce discomfort [19]. Total hip arthroplasty is recommended for patients who do not experience adequate pain reduction and improvements in physical function resulting from a combination of pharmacological and non-pharmacological treatment [20]. The procedure significantly improves patients’ quality of life, reduces their pain and dysfunction, and improves psychological comfort [21]. The most commonly used agents for postoperative pain relief are non-opioid analgesics (non-steroidal anti-inflammatory drugs, paracetamol, metamizole) and opioid analgesics combined with various methods of local anesthesia. Multimodal analgesia, which consists of administering drugs with different mechanisms of action to achieve a synergistic effect, is also successfully used [22]. The perception of postoperative pain may be influenced by preoperative or immediate postoperative discomfort, which predisposes the patient to develop excessive neural excitability and central sensitization. Another explanation may be intraoperative nerve damage. In addition, attention is paid to psychosocial and genetic factors predisposing to decreased pain excitability and the development of chronic postoperative pain [23].

In addition to hereditary disease entities impairing the degree of pain perception, one of the more interesting modulators affecting pain perception may be the polymorphism of the cannabinoid receptor type 1 (*CNR1*) gene on chromosome 6 (6q14-q15) [24]. Genetic variation in cannabinoid receptors affects osteoclasts, osteoblasts, bone formation, and bone mass [25,26]. The analgesic effect of cannabinoids involving the CB1 receptor is achieved by inhibiting GABA-ergic transmission in the periaqueductal gray (PAG) and ventral tegmental area (VTA) of the *medulla oblongata*, i.e., the structures responsible for the descending (antinociceptive) pathway of pain inhibition [27]. In vivo and in vitro electrophysiological studies have confirmed that CB1 activation by both endocannabinoids and exogenous cannabinoids affects serotonin output in the sutural nuclei [28].

The alternative splicing of CB1 mRNA produces six 5′ untranslated region (5′-UTR) splice variants. Five of them encode full-length CB1 protein, i.e., 472 amino acids. The alternative splicing in the exon 4 region leads to the deletion of 102–167 nts in the 5′ end of exon 4 or two different translation start sites are present from the 5′ end of exon 4. Authors [29,30,31] have demonstrated that the amino-terminal variants CB1 protein have reduced affinity for cannabinoid agonists and antagonists but not for the ligands. The knowledge of CB1 gene expression regulation is limited. CB1 receptor abundance and the endocannabinoid system’s function may change in response to altered CB1 gene expression in different developmental or disease conditions or in response to drug exposure. The CB1 mRNA transcription is malleable, and that fact may be exploited for therapeutic benefit [32]. One of the most frequently described variations within this gene is a microsatellite polymorphism of the number of three-nucleotide repeats (AAT), located in the 3′UTR non-translational region, more than 18,000 base pairs from exon 4 [33]. Due to their high degree of polymorphism, even and frequent distribution across the genome, microsatellites have become excellent genetic markers. The exact functional consequences of the (AAT)n polymorphism are not fully known. However, it is speculated that, similar to other microsatellite polymorphisms, it may promote the formation of a Z-DNA structure that affects gene expression. It is known that the concentration of nascent CB1 receptor protein is inversely proportional to the number of repeats in the gene [34]. Therefore, a higher number of repeats is more common; for instance, among Caucasians addicted to heroin, while a lower number of repeats appear to have a protective effect [35,36].

In most cases, surgical removal of the cause of peripheral nociceptor irritation leads to a reduction or cessation of pain. However, based on clinical observations, it is possible to distinguish a group of patients with higher pain intensity in whom surgical treatment does not lead to a satisfactory reduction of complaints. Therefore, it is worth noting the possible causes of individual differences in pain sensation as the main factor that worsens patients’ quality of life after hip arthroplasty. Since the polymorphism of the gene for the opioid receptor seems to show correlations with pain sensation [37], the aim of the study was to examine the associations between (AAT)n polymorphism of the *CNR1* gene encoding the cannabinoid CB1 receptor and the patient’s life quality after hip arthroplasty, with a particular focus on the degree of pain sensation, hip function, and the patient’s performance.

## 2. Materials and Methods

### 2.1. Characteristics of the Study Group

The study included 198 patients from the Department of Orthopedics, Traumatology and Musculoskeletal Oncology of the Pomeranian Medical University in Szczecin (Poland) who qualified for elective hip replacement due to degenerative changes. Of all the patients in the clinical follow-up, 64.6% were women (128 patients), and 35.4% were men (70 patients). The mean age was 67 years (±11.0 years). The inclusion criteria entailed patients referred to the Orthopedic Department due to osteoarthritic changes of the hip joints (on an elective basis), diagnosis of degenerative changes in the hip joints based on clinical examination and imaging—radiological examination (described as intermediate or advanced changes), age above 18 years, and written consent of the patient to participate in the study based on the “Patient Information” approved by the Local Bioethics Committee. The exclusion criteria entailed patients qualified for re-arthroplasty; osteoporosis that disqualifies the possibility of surgery, long-standing diabetes with features of polyneuropathy, diagnosed motor-sensory polyneuropathy, patients seeking disability benefits or ongoing litigation patients; mood disorders of depression-like nature (treated with thymoleptic drugs), addiction to opioid analgesics and anxiolytic drugs, and patients with psychoorganic syndrome (POS) (MMSE < 10 points). The Bioethics Committee approved of the planned study (No. BN-001/6/07 with further amendments).

The physical examination of the patients included a complete assessment of orthopedic status (body structure, posture, symmetry, and proportions), an examination of the spine, range of motion in the joints, an assessment of muscle strength, and neurological status. All patients participating in the study underwent cementless total arthroplasty from anterolateral access. Before surgery, on the first postoperative day, and after six weeks, each patient also underwent a follow-up X-ray of the pelvis with hip joints to assess the proper placement of the endoprosthesis. In the postoperative period, patients benefited from multimodal analgesic therapy in accordance with recommendations for postoperative pain management [38].

### 2.2. Assessment of Pain Sensation, Hip Function, and Degree of Disability

#### 2.2.1. The VAS Scale

The Visual Analogue Scale (VAS scale) allowed us to assess the pain intensity according to the increasing number of points on a scale from 0 to 10, i.e., 0 points mean no pain, while 10 points mean the most severe pain experienced in one’s life. In the study presented here, examinations using the VAS scale were performed 1.5 months (i.e., six weeks after the surgery), and six months after hip replacement.

#### 2.2.2. The Harris Hip Score

The Harris Hip Score (HHS) assesses the degree of pain, function, the degree of deformity, and range of motion of the operated joint (according to the scheme: Pain, Function, Deformity, Motion). This scale requires determining the severity of parameters including pain sensation (6 degrees), the distance possible to walk (4 degrees), daily activities such as putting on shoes or socks (3 degrees), ability to use public transportation (2 degrees), limping (4 degrees), use of orthopedic supplies—support by crutches or cane (6 degrees), ability to climb stairs (6 degrees), the comfort of sitting (3 degrees), limb length difference (in centimeters), and joint mobility (in degrees)—concerning flexion, extension, external and internal rotation, and inversion and adduction.

The results obtained using the HHS are assessed as inadequate with a score of less than 70. The range between 70 and 79 is average, 80–89 is good, and 90–99 is very good. According to the Harris scale, the maximum evaluation score is 100 points [39]. We used the HHS three times, i.e., one week after surgery (during a routine follow-up examination) and 1.5 months (i.e., six weeks), and six months after surgery.

#### 2.2.3. Oswestry Disability Questionnaire

The Oswestry Disability Questionnaire determines the patient’s degree of disability. Using this tool, the patient answers 10 questions about pain intensity and various spheres of life, including self-care, sleeping, sitting, standing, walking, lifting, traveling, social life, and work. The patients’ answers allowed us to assess their quality of life. The patient scores from 0 to 5 points for each answer, thus the maximum possible score is 50 points. A score of 0–4 points means no disability, 5–14 points—slight (mild) disability, and 15–24 points—moderate disability. A score of 25–34 points indicates severe disability, and a score above 35 points is total disability. Based on the Oswestry questionnaire, the Oswestry Disability Index (ODI) can be determined, which is calculated by dividing the sum of the Oswestry questionnaire scores by the number of questions on the questionnaire, which is then multiplied by 100%. The interpretation of the ODI distinguishes slight disability (10–20%), mild disability (21–40%), medium disability (41–60%), and severe disability (61–80%). Obtaining 81–100% indicates a very severe disability, which requires 24/7 third-party care. Testing with the ODI was carried out three times, i.e., one week after surgery (during a routine follow-up examination), after 1.5 months (i.e., after six weeks), and six months after surgery.

### 2.3. Genotyping

The day before surgery, a 2 mL whole blood sample for genetic testing was drawn. Genomic DNA was isolated using GeneMATRIX Quick Blood DNA Purification Kit (EURx, Gdańsk, Poland). Our study used a PCR genotyping assay to identify the (AAT)n triplet repeat polymorphism in the *CNR1* gene. PCR amplification was performed using primer forward: 5′-GCTGCTTCTGTTAACCCTGC-3′ and primer reverse: 5′-TACATCTCCGTGTGATGTTCC-3′ as described by Dawson [40]. PCR was carried out in 12 μL volumes containing 20 ng DNA; 1 X AmpliTaq360 buffer [Applied Biosystems, Waltham, MA, USA]; 200 μM dNTP mix [Applied Biosystems], 2 mM MgCl_2_ [Applied Biosystems]; 2 pmol fluorescent labeled primer forward; 2 pmol primer reverse and 0.6 U AmpliTaq 360 DNA polymerase [Applied Biosystems]. PCR was performed in a Mastercycler Gradient [Eppendorf] and included a three-minute initial denaturation at 94 °C followed by 26 cycles of the 20-second denaturation at 94 °C, primer annealing at 58 °C for 30 s, extension at 72 °C for 30 s, and a final extension at 72 °C for 30 min. Fluorescent analysis of PCR products on ABI PRISM^®^ 3100-Avant [Applied Biosystems] were used for length and allele determination. Additionally, (AAT)n repeat number was confirmed by sequencing in randomly selected homozygous samples using a standard sequencing protocol, performed on an ABI PRISM^®^ 3100-Avant [Applied Biosystems]. For the purpose of the analysis, alleles were subsequently classified as short (<11) or long alleles ≥ 11 (AAT)n, which is consistent with the research by Comings et al., [41].

### 2.4. Statistical Analysis

The accordance of genotype distributions with Hardy-Weinberg’s equilibrium was analyzed using the exact test. In most cases, the distributions of measurable parameters—pain perception, degree of disability, and hip function—were significantly different than normal (*p* < 0.05, Shapiro-Wilk test); thus, a non-parametric test was used in the calculations. Parameters and their changes were compared between genotypes using the Mann-Whitney test. The correlations between the number of AAT repeats in the shorter and longer allele, the average number of AAT repeats from two alleles, and measurable parameters were analyzed using Spearman’s rank correlation coefficient. Statistical calculations were performed using Statistica 10. In this case, *p* < 0.05 was used as the threshold for statistical significance.

## 3. Results

In the patients’ X-ray examinations, no complications such as endoprosthesis dislocation, endoprosthesis loosening, or peri-prosthesis fracture were found, which could have been the direct cause of chronic pain associated with the arthroplasty procedure. The patients were assessed for pain intensity using the VAS scale 1.5 months (6 weeks) and 6 months after surgery. Hip function using the HHS and the degree of disability using the ODI were also performed. The results of these measurements are shown in Table 1, Table 2 and Table 3.

The study evaluated the association of the AAT polymorphism of the cannabinoid receptor type 1 (*CNR1*) gene with pain intensity, hip function, and the degree of disability. The frequency of *CNR1* (AAT)n alleles in the study group is presented in Figure 1.

At the same time, Table 4 shows the frequency of *CNR1* genotypes (alleles classified as short or long, according to the determined number of AAT repeats).

Alleles with less than 11 AAT repeats were classified as short (S), while those with AAT number of repeats of 11 and above were described as long (L). Based on this classification, the patients were divided into three groups: those with 2 short alleles (S-S), those with 2 long alleles (L-L), and those with 1 short and 1 long allele (S-L). The distributions of *CNR1* genotypes were found to be consistent with Hardy-Weinberg’s equilibrium (*p* = 0.50). The study group’s short allele (S) frequency was 35%; here, there were no statistically significant differences in *CNR1* allele frequencies between genders. Similarly, the frequency of the three genotypes (S-S, S-L, L-L) of *CNR1* was similar in women and men—as shown in Table 5.

The study aimed to analyze the association of the cannabinoid CB1 receptor *CNR1* gene polymorphism with pain intensity, hip function, and the degree of disability in patients at defined intervals after surgery. There were no statistically significant correlations between the degree of pain sensation assessed according to the VAS scale and the L-L, S-L, and S-S genotypes at defined intervals, i.e., six weeks and six months after hip replacement)—Table 6.

There were also no statistically significant differences between hip function and the change (improvement) in hip function measured according to the Harris scale, L-L, S-L, and S-S genotypes at defined time intervals (i.e., one week, six weeks, and six months after the surgical treatment)—Table 7.

In addition, no statistically significant correlation was found between the degree of disability and the change in the degree of disability measured by the ODI, L-L, S-L, and S-S genotypes at defined intervals, i.e., one week, six weeks, and six months after the surgical treatment—Table 8.

There were no statistically significant correlations between the clinical symptoms and their changes, as measured by the VAS scale, Harris scale, Oswestry Disability Index, and the number of AAT repeats in the shorter allele and in the longer allele, as well as the average number of AATs from both alleles at defined time intervals (Table 9, Table 10 and Table 11).

The average number of AAT repeats for each patient was calculated based on the following formula: (number of AAT repeats in the shorter allele + number of AAT repeats in the longer allele/2).

## 4. Discussion

The endocannabinoid system is involved in several physiological and pathological processes; hence, attempts to use it for therapeutic purposes seem justified. Since the effect of this system, and especially the use of drugs affecting this system, may be subject to genetic variations, we chose to study one of the most frequently described variations within *CNR1* gene, i.e., a microsatellite polymorphism of the number of three-nucleotide repeats (AAT).

For the evaluation of the quality of life in patients after hip replacement, the Visual Analog Pain Scale (VAS) was used to assess pain, and the Harris Hip Score (HHS) and Oswestry Disability Index (ODI) were used to assess the performance of the patients after hip arthroplasty. Other authors have also used similar tools [42,43].

To analyze the effect of the polymorphism of the *CNR1* gene encoding the cannabinoid receptor (CB1) on the patient’s quality of life after hip replacement, the variable number of trinucleotide repeats (AAT)n within the *CNR1* gene encoding the CB1 receptor was determined in all patients. To identify (AAT)n, the method of Comings et al. [41] was used, with an in-house modification in the form of labeling the sense primer with a fluorescent dye. *CNR1* alleles were subsequently classified into two groups according to the criterion of the number of repeats of the AAT sequence, distinguishing the short allele (S)—containing less than 11 repeats, and the long allele (L)—containing 11 and more repeats. This was used as the basis for a division of patients into 3 genotypes (S-S, S-L, L-L), whose frequency distribution among the patients qualified for our study was similar to the results obtained by other authors [44]. A correlation analysis was then performed to assess the association between clinical factors and the frequency of the short and long alleles. 

In our study, we did not find an association between the (AAT)n polymorphism of the *CNR1* gene encoding cannabinoid receptor type 1 and the pain sensation of the patients after hip arthroplasty. Other authors have also looked at the role of genetic variation in the CB1 receptor in pain perception. The Brazilian analysis [45] found 30 genes related to fibromyalgia (FM) influencing the symptoms of this disease. That study suggested that rs6454674, rs1078602, and rs10485171 in the *CNR1* gene may be associated with FM, obesity, irritable bowel syndrome, migraine, and post-traumatic stress disorder. On the other hand, Gerra et al., [46] found no significant associations using the family-based analysis or the SNPs. The authors suggested that patients with FM without depression and those with FM and depression show a significant difference in the genotypic distribution related to SNP rs6454674 in the cannabinoid receptor 1 gene (*CNR1*). This indicates that FM is not a homogeneous disorder. Spanish researchers [47] analyzed a single nucleotide polymorphism (*CNR1* G1359A, rs12720071) in a similar area to the AAT microsatellite polymorphism, i.e., the 3′UTR of exon 4. The pain threshold and pain tolerance were determined based on the cold-pressor test, which involved immersion of the non-dominant hand in 4 °C water by young, healthy participants. Perceived pain was also assessed using the VAS scale. However, none of the variables were found to be statistically significantly associated with the studied polymorphism, regardless of the participant’s assessed mood. Furthermore, in another study [48], *CNR1* mRNA expression did not differ among chronic low back pain patients. Researchers from Japan [49] performed an association study between (ATT)n repeats in the *CNR1* gene, peripheral pain sensation, and analgesia-related features in postoperative pain management in patients who underwent open abdominal or orthognathic cosmetic surgery. In that study, no statistically significant associations were found between (ATT)n repeats in the *CNR1* gene and peripheral pain sensation. However, short tandem repeats in the *CNR1* gene were associated with the frequency of fentanyl use, fentanyl dose, and the VAS scores 3 h after orthognathic cosmetic surgery. Similar observations were obtained in the study of patients with irritable bowel syndrome, in which no association between the (ATT)n repeats of the *CNR1* gene and pain sensation was found [50]. Based on our analysis of the literature, we believe that pain perception is influenced by the complicity of many genetic and environmental factors. It is, therefore, difficult to determine the impact of individual components.

The intensity of pain perception assessed using the visual analogue scale (VAS) was not found to be statistically significantly correlated with the (AAT)n polymorphism of the CB1 cannabinoid receptor *CNR1* gene in patients after hip arthroplasty. As measured by the Harris scale, there was also no statistically significant relationship between the CB1 receptor *CNR1* gene polymorphism and hip function. Furthermore, there was no statistically significant correlation between the degree of disability and the change in the degree of disability as measured by the ODI. However, it is important to note that this has been groundbreaking research and that more research is needed. The first observations linking the endocannabinoid system to postoperative pain after hip replacement and to functional disability in arthrosis patients were only made a few years ago [51]. Studies to date have mainly focused on using cannabinoids in postoperative multimodal pain management in surgical patients [52,53,54]. Still, they have not yielded a conclusive answer to consider including cannabinoids in the standard of postoperative treatment in orthopedics. To date, arthroplasty has been the main type of treatment, whilst cannabinoid application is still discussed.

Some researchers have shown that single-stranded miRNAs bound in the RNA-induced risk complex (RISC) silencing complex bind to non-coding 3′-untranslated regions (3′UTR) of mRNA, leading to decreased translation, increased transcript degradation, or both [55]. Therefore, polymorphisms in the 3′-untranslated region (3′UTR) may lead to gene expression changes by modifying the attachment sites of the RISC complex. Unfortunately, it is impossible to be certain whether the results presented in our study were affected by this issue. Another step in the future study should be to quantitatively analyze the nascent CB1 protein and investigate the effect of the microsatellite (AAT)n polymorphism on translation and the protein expression of the cannabinoid system using proteomics tools. Next, there will be more premises to evaluate the correlation between the amount of protein and the parameters studied in patients after hip arthroplasty. Learning about the importance of factors influencing the function of the cannabinoid system from a therapeutic perspective seems important, especially when the predominant symptom of hip osteoarthritis is chronic pain of a receptor nature. Next, the studies conducted may be helpful in choosing the method and intensity of treatment for patients. From a broad perspective, as other authors [56] also indicate, studying other genes related to pain sensation is worthwhile also. Genes containing associated variants include the transient receptor potential A subtype 1 gene (TRPA1), the catechol-o-methyltransferase gene (COMT), the fatty acid amide hydrolase gene (FAAH), and the endogenous opioid receptors (such as OprM and OprK).

## 5. Conclusions

In conclusion, the aim of our study was to find a potential association between one of the genetic factors—(ATT)n polymorphism—and the perception of pain, hip function, and performance in patients after hip arthroplasty. The analysis of the results showed that there was no association between the length of the allele of the CB1 cannabinoid receptor *CNR1* gene and the degree of pain, hip function, and change in disability in patients who underwent elective hip arthroplasty. However, due to the relatively small group size, larger studies are needed in order to confirm the value of the present study observations, as the research on the cannabinoid system may open new perspectives for individualizing therapy and evaluating the effectiveness of treatment for patients with hip osteoarthritis. Considering the undeniable role of the cannabinoid system in nociception, further research in this area would be advisable.

## Figures and Tables

**Figure 1 genes-13-02046-f001:**
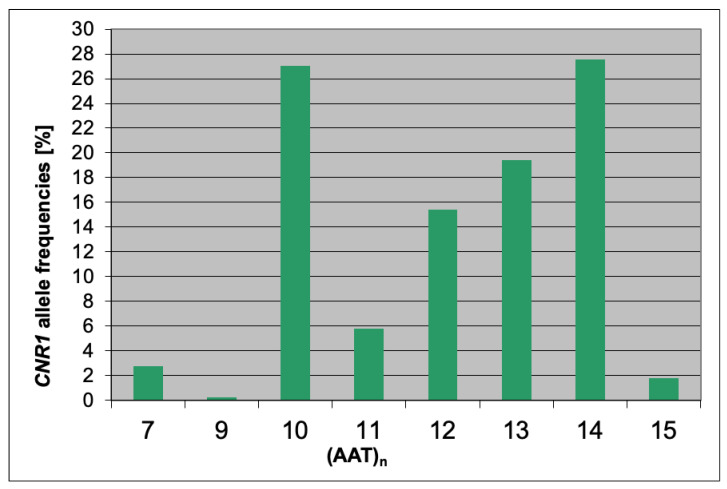
Frequency of *CNR1* (AAT)n allele frequencies in patients after hip arthroplasty.

**Table 1 genes-13-02046-t001:** Analysis of the degree of pain experienced by patients after hip arthroplasty at defined intervals.

Scale	Mean ± SD	Median	Lower Quartile	Upper Quartile	(Min–Max)
VAS (after 1.5 month)	3.0 ± 1.1	3	2	4	(0–7)
VAS (after 6 months)	1.5 ± 1.7	1	0	2	(0–9)

**Table 2 genes-13-02046-t002:** Evaluation of patients‘ hip function after hip arthroplasty at defined intervals.

Scale	Mean ± SD	Median	Lower Quartile	Upper Quartile	(Min–Max)
HHS (after 1 week)	34.9 ± 12.7	37	26	43	(0–70)
HHS (after 1.5 month)	69.8 ± 11.7	71	64	77	(17–96)
HHS (after 6 months)	86.8 ± 11.5	90	83	95	(32–99)
Δ HHS (1.5 month—1 week)	34.9 ± 15.1	33.5	26	44	(−14–78)
Δ HHS (6 months—1 week)	51.9 ± 14.4	52	43	61	(10–94)

HHS—points obtained on the Harris scale. **Δ** HHS—delta HHS—the difference between HHS after 1.5 or 6 months and HHS after 1 week.

**Table 3 genes-13-02046-t003:** Disability analysis of patients after hip arthroplasty at defined intervals.

Scale	Mean ± SD	Median	Lower Quartile	Upper Quartile	(Min–Max)
ODI (after 1 week)	52.2 ± 16.1	50	40	64	(18–96)
ODI (after 1.5 month)	18.2 ± 10.9	16	12	22	(0–89)
ODI (after 6 months)	7.1 ± 10.1	4	0	9	(0–58)
Δ ODI (1.5 month—1 week)	−34.0 ± 17.7	−30.5	−46	−23	(−82–13)
R%Δ ODI (1.5 month—1 week)	−62.7 ± 20.9	−64.7	−77	−53	(−100–50)
Δ ODI (6 months—1 week)	−45.1 ± 18.2	−44	−58	−33	(−95–28)
R%Δ ODI (6 months—1 week)	−85.5 ± 25.5	−93.9	−100	−80	(−100–156)

ODI—Oswestry Disability Index. Δ ODI—delta ODI—the difference between ODI after 1.5 or 6 months and ODI after 1 week. R%Δ—relative change of parameter defined as 100* (ODI after 1.5 or 6 months—ODI after 1 week)/ODI after 1 week.

**Table 4 genes-13-02046-t004:** Frequency distribution of short and long CNR1 genotypes classified by AAT number of repeats in the study group.

Genotypes	Female		Male		*p* ^a^
	N	%	N	%	
S-S	11	8.59%	9	12.86%	0.51
S-L	54	42.19%	25	35.71%	
L-L	63	49.22%	36	51.43%	

^a^ χ^2^ test. HWE *p* = 0.50. MAF (S): 0.35. MAF—frequency of rare allele. S—short allele (AAT)n < 11. L—long allele (AAT)n ≥ 11.

**Table 5 genes-13-02046-t005:** Frequency of *CNR1* genotypes according to gender.

Genotypes	Female		Male		*p* ^a^
	N	%	N	%	
S-S	11	8.59%	9	12.86%	0.51
S-L	54	42.19%	25	35.71%	
L-L	63	49.22%	36	51.43%	

^a^ χ^2^ test. S—short allele (AAT)n < 11. L—long allele (AAT)n ≥ 11.

**Table 6 genes-13-02046-t006:** Association of *CNR1* genotype with pain intensity in patients at defined intervals.

Scale	Genotypes					S-S vs. S-L + L-L	S-S + S-L vs. L-L	S-S vs. L-L	S-S vs. S-L	L-L vs. S-L
	S-S (*n* = 20)	S-L (*n* = 79)	L-L (*n* = 99)	S-S + S-L (*n* = 99)	S-L + L-L (*n* = 178)	*p* ^&^				
	Mean ± SD									
VAS (after 1.5 month)	2.8 ± 1.1	3.1 ± 1.1	3.0 ± 1.0	3.0 ± 1.1	3.0 ± 1.1	0.21	0.56	0.19	0.27	0.88
VAS (after 6 months)	1.0 ± 1.0	1.4 ± 1.7	1.6 ± 1.8	1.3 ± 1.5	1.5 ± 1.7	0.34	0.29	0.26	0.52	0.44

^&^ U Mann-Whitney test; VAS—scale VAS; S—short allele (AAT)n < 11; L—long allele (AAT)n ≥ 11.

**Table 7 genes-13-02046-t007:** Association of *CNR1* genotype with hip function assessment in patients at defined intervals.

Scale	Genotypes					S-S vs. S-L + L-L	S-S + S-L vs. L-L	S-S vs. L-L	S-S vs. S-L	L-L vs. S-L
	S-S (*n* = 20)	S-L (*n* = 79)	L-L (*n* = 99)	S-S + S-L (*n* = 99)	S-L + L-L (*n* = 178)	*p* ^&^				
	Mean ± SD									
HHS (after 1 week)	37.2 ± 9.8	35.4 ± 12.4	34.0 ± 13.4	35.8 ± 11.9	34.6 ± 12.9	0.38	0.36	0.31	0.55	0.50
HHS (after 1.5 month)	71.3 ± 11.7	70.2 ± 10.1	69.2 ± 12.9	70.4 ± 10.4	69.7 ± 11.7	0.34	0.88	0.41	0.30	0.87
HHS (after 6 months)	90.7 ± 4.5	87.3 ± 10.1	85.7 ± 13.2	88.0 ± 9.4	86.4 ± 11.9	0.24	0.55	0.25	0.27	0.81
Δ HHS (1.5 month—1 week)	34.1 ± 14.8	34.8 ± 14.1	35.3 ± 16.1	34.6 ± 14.2	35.1 ± 15.2	0.99	0.79	0.92	0.92	0.79
Δ HHS (6 months—1 week)	53.5 ± 9.2	51.8 ± 14.4	51.7 ± 15.3	52.2 ± 13.5	51.8 ± 14.9	0.51	0.91	0.63	0.43	0.74

^&^ U Mann-Whitney test; HHS—points obtained on the Harris scale; Δ HHS—delta HHS—the difference between HHS after 1.5 or 6 months and HHS after 1 week; S—short allele (AAT)n < 11; L—long allele (AAT)n ≥ 11.

**Table 8 genes-13-02046-t008:** Association of *CNR1* genotype with patients’ degree of disability at defined intervals.

Scale	Genotypes					S-S vs. S-L + L-L	S-S + S-L vs. L-L	S-S vs. L-L	S-S vs. S-L	L-L vs. S-L
	S-S (*n* = 20)	S-L (*n* = 79)	L-L (*n* = 99)	S-S + S-L (*n* = 99)	S-L + L-L (*n* = 178)	*p* ^&^				
	Mean ± SD									
ODI (after 1 week)	54.5 ± 15.0	52.0 ± 16.6	51.9 ± 16.1	52.5 ± 16.2	51.9 ± 16.2	0.40	0.94	0.46	0.37	0.83
ODI (after 1.5 month)	15.7 ± 8.0	18.2 ± 10.4	18.7 ± 11.8	17.7 ± 10.0	18.5 ± 11.2	0.21	0.88	0.25	0.23	0.76
ODI (after 6 months)	3.9 ± 3.7	6.7 ± 9.4	8.0 ± 11.4	6.1 ± 8.6	7.4 ± 10.6	0.41	0.39	0.32	0.59	0.55
Δ ODI (1.5 month—1 week)	−38.8 ± 17.1	−33.7 ± 17.6	−33.2 ± 17.9	−34.7 ± 17.5	−33.4 ± 17.7	0.16	0.71	0.20	0.17	0.93
R%Δ ODI (1.5 month—1 week)	−69.4 ± 16.4	−62.7 ± 18.5	−61.4 ± 23.3	−64.0 ± 18.2	−62.0 ± 21.3	0.11	0.72	0.14	0.11	0.86
Δ ODI (6 months—1 week)	−50.6 ± 15.8	−45.3 ± 18.2	−43.8 ± 18.6	−46.4 ± 17.8	−44.5 ± 18.4	0.11	0.55	0.12	0.14	0.96
R%Δ ODI (6 months—1 week)	−91.9 ± 9.7	−86.9 ± 18.2	−83.1 ± 31.8	−87.9 ± 16.9	−84.8 ± 26.6	0.33	0.45	0.33	0.40	0.63

^&^ U Mann-Whitney test; ODI—Oswestry Disability Index; Δ ODI—delta ODI—the difference between ODI after 1.5 or 6 months and ODI after 1 week; R%Δ—relative change of parameter defined as 100* (ODI after 1.5 or 6 months—ODI after 1 week)/ODI after 1 week; S—short allele (AAT)n < 11, L—long allele (AAT)n ≥ 11.

**Table 9 genes-13-02046-t009:** The correlations between the number of AAT repeats in the *CNR1* gene in the shorter allele, pain intensity, hip function, and the degree of disability in patients at defined intervals.

Scale/Questionnaire	Number of AAT Repeats in the Shorter Allele	
	R_s_	*p*
VAS (after 1.5 month)	0.03	0.63
VAS (after 6 months)	0.10	0.16
HHS (after 1 week)	−0.04	0.59
HHS (after 1.5 month)	0.01	0.92
HHS (after 6 months)	−0.04	0.61
Δ HHS (1.5 month—1 week)	0.02	0.83
Δ HHS (6 months—1 week)	−0.01	0.92
ODI (after 1 week)	−0.04	0.56
ODI (after 1.5 month)	−0.01	0.87
ODI (after 6 months)	0.04	0.54
Δ ODI (1.5 month—1 week)	0.03	0.63
R%Δ ODI (1.5 month—1 week)	0.01	0.89
Δ ODI (6 months—1 week)	0.07	0.36
R%Δ ODI (6 months—1 week)	0.04	0.54

R_S_—Spearman’s rank correlation coefficient; VAS—VAS scale; HHS—scores obtained on the Harris scale; ODI—Oswestry Disability Index; Δ HHS—delta HHS—the difference between HHS after 1.5 or 6 months and HHS after 1 week; Δ ODI—delta ODI—the difference between ODI after 1.5 or 6 months and ODI after 1 week; R%Δ—relative change of parameter defined as 100* (ODI after 1.5 or 6 months—ODI after 1 week)/ODI after 1 week.

**Table 10 genes-13-02046-t010:** The correlations between the number of AAT *CNR1* repeats in the longer allele, pain intensity, hip function, and the degree of disability in patients at defined intervals.

Scale/Questionnaire	Number of AAT Repeats in the Longer Allele	
	R_s_	*p*
VAS (after 1.5 month)	0.08	0.27
VAS (after 6 months)	0.10	0.18
HHS (after 1 week)	−0.11	0.11
HHS (after 1.5 month)	−0.03	0.70
HHS (after 6 months)	−0.08	0.27
Δ HHS (1.5 month—1 week)	0.04	0.54
Δ HHS (6 months—1 week)	−0.001	0.99
ODI (after 1 week)	−0.07	0.36
ODI (after 1.5 month)	−0.01	0.89
ODI (after 6 months)	−0.01	0.94
Δ ODI (1.5 month—1 week)	0.08	0.24
R%Δ ODI (1.5 month—1 week)	0.07	0.35
Δ ODI (6 months—1 week)	0.12	0.10
R%Δ ODI (6 months—1 week)	0.01	0.85

R_S_—Spearman’s rank correlation coefficient; VAS—VAS scale; HHS—scores obtained on the Harris scale; ODI—Oswestry index; Δ HHS—delta HHS—the difference between HHS after 1.5 or 6 months and HHS after 1 week; Δ ODI—delta ODI—difference between ODI after 1.5 or 6 months and ODI after 1 week; R%Δ—relative change of parameter defined as 100* (ODI after 1.5 or 6 months—ODI after 1 week)/ODI after 1 week.

**Table 11 genes-13-02046-t011:** The correlations between the average number of AAT repeats in the *CNR1* gene from both alleles, pain intensity, hip function, and the degree of disability in patients at defined intervals.

Scale/Questionnaire	Average Number of AATs from both Alleles ^a^	
	R_s_	*p*
VAS (after 1.5 month)	0.06	0.41
VAS (after 6 months)	0.12	0.099
HHS (after 1 week)	−0.08	0.24
HHS (after 1.5 month)	−0.01	0.91
HHS (after 6 months)	−0.08	0.25
Δ HHS (1.5 month—1 week)	0.03	0.64
Δ HHS (6 months—1 week)	−0.01	0.84
ODI (after 1 week)	−0.06	0.38
ODI (after 1.5 month)	−0.01	0.91
ODI (after 6 months)	0.03	0.66
Δ ODI (1.5 month—1 week)	0.07	0.34
R%Δ ODI (1.5 month—1 week)	0.04	0.56
Δ ODI (6 months—1 week)	0.10	0.15
R%Δ ODI (6 months—1 week)	0.04	0.59

^a^ (number of AAT repeats in the shorter allele + number of AAT repeats in the longer allele)/2; R_S_—Spearman’s rank correlation coefficient; VAS—VAS scale; HHS—scores obtained on the Harris scale; ODI—Oswestry Disability Index; Δ HHS—delta HHS—the difference between HHS after 1.5 or 6 months and HHS after 1 week; Δ ODI—delta ODI—the difference between ODI after 1.5 or 6 months and ODI after 1 week; R%Δ—relative change of parameter defined as 100* (ODI after 1.5 or 6 months—ODI after 1 week)/ODI after 1 week.

## Data Availability

The datasets generated and/or analyzed during the current study are available from the corresponding author on reasonable request.

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
