# Peer review of "Lack of Association between (AAT)n Polymorphism of the CNR1 Gene Encoding the Cannabinoid Receptor (CB1) and Patient’s Quality of Life"

_genes, 2022, doi:10.3390/genes13112046_

Round 1

Reviewer 1 Report

In the current paper the authors describe the important role of CNR1 gene polymorphs - cannabinoid type 1 receptor. Understanding how this polymorphism relates to the variability of the body's response to pain is quite essential.

Therefore, it is worth emphasizing that the topic undertaken by the authors is very innovative and extremely important in terms of cognition and in diagnostics and targeted therapy.

This work was methodically well planned, and interesting results were obtained and well interpreted by the authors.

In the presented study, no statistically significant relationships were found between CNR1 genotypes and the intensity of pain sensation, and the degree of disability in the studied groups.

After analyzing the work, the question arises: in the case of the gene encoding the type 1 cannabinoid receptor, can there be changes in the processes of translation or an increase in transcript degradation?

Author Response

We are grateful to the reviewers for the valuable comments regarding the manuscript ID: genes-1986464 entitled "Association between (AAT)n polymorphism of the CNR1 gene encoding the cannabinoid receptor (CB1) and patient’s quality of life" submitted by Machoy and Rac et al. to “Genes”. The point-by-point responses to the reviewers` comments are presented below. The changes introduced to the manuscript are marked red.

According to the reviewer`s suggestion, we added the sentences to the “Introduction” section with the new corresponding reference [29-32]:

The alternative splicing of CB1 mRNA produces six 5′ untranslated region (5′-UTR) splice variants. Five of them encode full-length CB1 protein, i.e., 472 amino acids. The alternative splicing in the exon 4 region leads to the deletion of 102-167 nts in the 5′ end of exon 4 or two different translation start sites are present from the 5′ end of exon 4. Authors [Straiker A., Wagner-Miller J., Hutchens J., Mackie K. Differential signaling in human cannabinoid CB1 receptors and their splice variants in autaptic hippocampal neurons. Br. J. Pharmacol. 2012, 165: 2660–2671 doi: 10.1111/j.1476-5381.2011.01744.x., Ryberg E., Vu H.K., Larsson N., Groblewski T., Hjorth S., Elebring T., Sjögren S., Greasley P.J. Identification and characterisation of a novel splice variant of the human CB1receptor. FEBS Lett. 2005, 579: 259–264 doi: 10.1016/j.febslet.2004.11.085., Xiao J.C., Jewell J.P., Lin L.S., Hagmann W.K., Fong T.M., Shen C.P. Similar in vitro pharmacology of human cannabinoid CB1 receptor variants expressed in CHO cells. Brain Res. 2008, 1238: 36–43 doi: 10.1016/j.brainres.2008.08.027.] have demonstrated that the amino-terminal variants CB1 protein have reduced affinity for cannabinoid agonists and antagonists but not for the ligands. The knowledge of CB1 gene expression regulation is limited. CB1 receptor abundance and the endocannabinoid system's function may change in response to altered CB1 gene expression in different developmental or disease conditions or in response to drug exposure. The CB1 mRNA transcription is malleable, and that fact may be exploited for therapeutic benefit [Laprairie R.B., Kelly M.E.M., Denovan-Wright E.M. The dynamic nature of type 1 cannabinoid receptor (CB1) gene transcription. Br. J. Pharmacol. 2012 167(8): 1583–1595 doi: 10.1111/j.1476-5381.2012.02175.x].

Reviewer 2 Report

This manuscript presents an association analysis of the (AAT)n repeat polymorphism in the 3’ UTR of the endocannabinoid receptor gene CB1, with pain tolerance and quality of life of post operative osteoarthritis patients. The authors have used the visual analog scale for pain quantification, Harris hip score for hip functionality quantification, and a Oswestry questionnaire to measure post-operative quality of life. No significant association was found between (AAT)n repeat polymorphism and the three measures.

This paper can be improved by broadening its scope, and some additional analyses:

  • Previous studies have implicated variants in other genes (such as TRPA1, COMT, and FAAH) related to the cannabinoid pathway to be associated with pain tolerance (PMID: 30468309). Including those variants in genotyping assays and either confirming or refuting the associations will be a significant leap towards understanding these pathways.
  • Studies have implicated that endogenous opioid receptors may play a role in pain tolerance as well (PMID: 30468309). Including variants from these genes (such as OPRM1 and OPRK1) may reveal interesting associations as well.
  • If no significant associations are found, the title should be changed to “No Association between (AAT)n polymorphism of the CNR1 gene and patient’s quality of life”.

Author Response

We are grateful to the reviewers for the valuable comments regarding the manuscript ID: genes-1986464 entitled "Association between (AAT)n polymorphism of the CNR1 gene encoding the cannabinoid receptor (CB1) and patient’s quality of life" submitted by Machoy and Rac et al. to “Genes”. The point-by-point responses to the reviewers` comments are presented below. The changes introduced to the manuscript are marked red.

R2. This paper can be improved by broadening its scope, and some additional analyses:

  1. Previous studies have implicated that variants in other genes (such as TRPA1, COMT, and FAAH) related to the cannabinoid pathway are associated with pain tolerance (PMID: 30468309). Including those variants in genotyping assays and either confirming or refuting the associations will be a significant leap towards understanding these pathways.

Studies have implicated that endogenous opioid receptors may also play a role in pain tolerance (PMID: 30468309). Including variants from these genes (such as OPRM1 and OPRK1) may also reveal interesting associations.

Unfortunately, we have yet to study the association of other gene variants and their influence on pain tolerance. According to the reviewer`s suggestion, we added the sentences to the end of the “Discussion” section with new corresponding reference [56]:

From a broad perspective,  as other authors [Patanwala A.E., Norwood C., Steiner H., Morrison D., Li M., Walsh K., Martinez M., Baker S.E., Snyder E.M., Karnes J.H., Psychological and genetic predictors of pain tolerance. Clin. Transl. Sci. 2019, 12(2): 189-195  doi: 10.1111/cts.12605] also indicate, studying other genes related to pain sensation is worthwhile also. Genes containing associated variants include the transient receptor potential A subtype 1 gene (TRPA1), the catechol‐o‐methyltransferase gene (COMT), the fatty acid amide hydrolase gene (FAAH), and the endogenous opioid receptors (such as OprM and OprK).

  1. If no significant associations are found, the title should be changed to “No Association between (AAT)n polymorphism of the CNR1 gene and patient’s quality of life”.

According to the reviewer’s suggestion, we modified the title to: “Lack of association between (AAT)n polymorphism of the CNR1 gene encoding the cannabinoid receptor (CB1) and patient’s quality of life”.
